# "*If I tell you my problems, how will you perceive me*?": A qualitative study of mental health knowledge, barriers, and opportunities for care among Kenyan adolescents during COVID-19

Vincent Nyongesa[1], Joseph Kathono[2], Shillah Mwavua[2,3], Darius Nyamai[4], Obadia Yator[5], Ian Kanyanya[6], Nabila Amin[7], Simon Njuguna[7,8], Jill Ahs[9], Brandon Kohrt[10], Bruce Chorpita[11], Beatrice Madeghe[5], Priscilla Idele[12], Liliana Carvajal Velez[9,13], Manasi Kumar[14]*

1 Department of Psychiatry, University of Nairobi, Nairobi, Kenya, 2 Department of Preventive and Promotive Health, Nairobi County Government, Nairobi, Kenya, 3 Department of Clinical Developmental Psychology, Vrije University, Amsterdam, Netherlands, 4 Nairobi County Government, Nairobi, Kenya, 5 University of Nairobi Infectious and Tropical Disease (UNITID), University of Nairobi, Nairobi, Kenya, 6 Department of Mental Health, Kenyatta National Hospital, Nairobi, Kenya, 7 Mathari National Teaching and Referral Hospital, Nairobi, Kenya, 8 Division of Mental Health, Ministry of Health, Nairobi, Kenya, 9 Department of Global Public Health, Karolinska Institutet, Stockholm, Sweden, 10 Department of Psychiatry and Behavioral Health, Center for Global Mental Health Equity, The George Washington University, Washington, DC, United States of America, 11 University of California, Los Angeles, California, United States of America, 12 Data and Analytics Branch United Nations Population Fund (UNFPA), New York, New York, United States of America, 13 United Nations Children's Fund, New York, New York, United States of America, 14 Institute for Excellence in Health Equity, New York University School of Medicine, New York, New York, United States of America

* manasi.kumar@nyulangone.org

## Abstract

### Introduction

The majority of the world's adolescents live in low-and middle-income countries (LMICs). However, there is a dearth of knowledge about adolescents' perspectives on mental health and sources of distress there. Therefore, a qualitative study of adolescents' and caregivers' beliefs and experiences related to mental illness was conducted in Nairobi, Kenya.

### Materials and methods

Six focus group discussions were conducted with 46 participants at two peri-urban settlements in Nairobi from November to December 2020. Using a two-step analytic process, we generated core themes, and the study team reviewed the transcripts and triangulated the themes.

---

**Data availability statement:** The data supporting this study cannot be shared publicly due to the potential presence of sensitive identifying information. However, the data are available from the KNH/UON Institutional Data Access / Ethics Committee (https://erc.uonbi.ac.ke/) for researchers who meet the necessary criteria for accessing confidential data. Inquiries regarding data access can be directed to the committee via email at uonknh_erc@uonbi.ac.ke.

**Funding:** Research reported in this publication was supported by the Fogarty International Center of the National Institutes of Health under Award Number K43TW010716, which also supported the contributions of MK to this work. The content is solely the responsibility of the authors and does not necessarily represent the official views of the National Institutes of Health. The qualitative inquiry is part of an embedded study protocol for the Measurement of Mental Health of Adolescents at population level (MMAP) led by UNICEF and the inquiry reported here was partially supported by UNICEF Innocenti office. The funders had no role in study design, data collection and analysis, decision to publish, or preparation of the manuscript.

**Competing interests:** The authors have declared that no competing interests exist.

## Results

Themes include knowledge about mental health and illness, triggers of psychological disturbances, attitudes towards mental illness, practices adopted to strengthen mental health, barriers to strengthening mental health among young people, emerging needs around caregiver mental health and parenting, and community recommendations on interventions. Adolescents had limited knowledge of specific mental illnesses but articulated triggers, stresses, and challenges they face in daily life in an in-depth manner. Caregivers demonstrated a breadth of knowledge and understanding of adolescent mental health.

## Conclusion

Adolescents and their caregivers face tremendous stresses in the Nairobi settlements. Mental health literacy is limited, but adolescents and caregivers are ready to embrace mental health services. Reducing stigma and access to youth-friendly services are crucial to expanding service engagement.

## Introduction

Mental health problems in adolescents and youth are common across the world, with an estimated 15% of adolescents in sub-Saharan Africa living with common mental conditions such as mood, anxiety, and stress disorders [1]. The burden for those living in low- and middle-income countries (LMICs), where 90% of the world's adolescents live [2], is exacerbated by a lack of access to care in these settings [3]. Mental health attitudes and barriers to the uptake of services and mitigation strategies vary from one sociocultural context to another [4]. These cultural differences co-occur with socioeconomic variations. In Kenya, poor health care infrastructure, poorly trained and empowered health workers, and lack of preventive and integrated mental health services are known barriers that come in the way of ensuring youth well-being [5].

Young people become particularly vulnerable from middle childhood years onwards to a sequela of psychosocial risks that can trigger mental, neurological, and substance use disorders [6]. Schooling, peer relationships, and the opportunity to engage in one's own networks are known to be protective factors in buffering adolescent and youth mental health [7].

The COVID-19 pandemic led to dramatic essential service disruptions, including schooling, health, and social services in Kenya, resulting in a high burden of additional needs impacting young people. These include an unmet need for psychological first aid and impaired access to mental health care and psychosocial support [8,9]. Systematic data on the mental health impact of COVID-19 on young people in such settings is still lacking [10]. However, research suggests that the COVID-19 pandemic's social and economic impact on adolescents and young adults in Nairobi, Kenya, is highly gendered, with young women and girls most affected [11,12]. Disproportionate risks and impacts shouldered by adolescent girls and households during the

COVID-19 pandemic are associated with increased risk for mental health problems, including the stress of food insecurity, risk of household violence, and forgoing necessary medical care [13].

A study in Nairobi among youth seeking mental health services at the Kenyatta National Referral Hospital found that they preferred to be offered both medication as well as psychosocial interventions over other forms of care. It was also found that young people were not inclined to believe in supernatural or traditional beliefs around mental illnesses; however, they did think stigma and family stressors were a considerable source of distress [14], often exacerbating mental distress.

As adolescents face a high prevalence of mental disorders coupled with limited access to care, delay in seeking care, or not seeking it altogether, a more understanding of adolescents' perceptions and knowledge about mental health and illness is needed. Identifying the barriers that prevent adolescents from seeking care for mental disorders and modifiable factors to reduce adolescents' risk and increase resilience is essential. This study aimed to explore adolescents' and caregivers' knowledge, perceptions, and insights about adolescent mental health living in two low-income settings in Nairobi.

## Methods

### Study design

A qualitative approach leveraging lived experiences was thought would capture perspectives, experiences, and individuals' reflections on adolescent mental health. Focus group discussions were conducted to investigate the complex themes surrounding adolescent mental health, as this method allows for peer group dynamic stimulation of discussions, enabling a secure environment for participants to convey a genuine picture of the subject [15]. A focus group discussion approach to explore mental health needs and concerns enabled an in-depth understanding of the knowledge gap, attitudes, and practices as the participants shared lived experiences. It allowed the facilitators to take note of nuances arising from the discussions and supported by the rich verbatim text data for analysis.

### Settings

**Kariobangi north and Kangemi health centers.** Kariobangi and Kangemi health centers [16] are level three facilities under the Nairobi County. Level three facilities include health centers, maternity homes, and sub-district hospitals. A level two facility is an equivalent to dispensaries, while level one is the basic level at the community health unit mandated to provide community health services [17]. Kariobangi health center is located in a low-income residential area in the northeastern part of Nairobi, Kenya. It consists of both lower middle class and slum-type with 18,903 residents [18]. On the other hand, Kangemi health center is located in a slum in Nairobi city within a small valley on the city's outskirts with 116,710 residents [18].

**Participants.** Thirty adolescent participants ages 10–19 years old and 16 caregivers of adolescents ages 10–14 years were recruited [19] by trained community health volunteers, who administered informed consent and assent in advance of the focus group discussions (FGDs) which were held at Kangemi and Kariobangi primary health care facilities. Only caregivers of adolescents ages 10–14 years were invited to the FGD for proxy reporting as opposed to those of older adolescents who were considered mature enough to talk for themselves. A total of six FGDs were conducted in November and December 2020 (See Table 1). We ensured that saturation of theme was achieved and hence there were no emerging themes by the time we completed the sixth FGD [20]. It has been recommended that about four focus groups were sufficient to identify a range of new issues (code saturation), but more groups were needed to fully understand these issues (meaning saturation) [19].

### Ethical approval

The study received ethical approval from the Institutional Ethics review committee of Kenyatta National Hospital and the University of Nairobi (No. P694/09/2018) and a research permit from Kenya National Commission for Science, Technology

**Table 1. Summary of FGD participants.**

| FGD set | Site | Cohort | N |
|---|---|---|---|
| First FGD | Kangemi health center | Girls 10–14 years | 8 |
| Second FGD | Kariobangi health center | Boys 10–14 years | 8 |
| Third FGD | Kariobangi health center | Girls 15–19 years | 8 |
| Fourth FGD | Kangemi health center | Boys 15–19 years | 6 |
| Fifth FGD | Kariobangi health center | Caregivers | 8 |
| Sixth FGD | Kangemi health center | Caregivers | 8 |

and Innovation (NACOSTI/P/19/77705/28063). Permission was also sought from these study sites through the Nairobi County Directorate of Health (Approval no. CMO/NRB/OPR/VOL1/2019/04). This study was nested under MK's work on Implementing mental health interventions for pregnant adolescents in primary care LMIC settings (INSPIRE) study. Written consent was obtained from participants, and informed assent was followed by parental/caregiver consent for participants who were below 18 years old. Participants were assigned unique identifiers to protect their identities, collected data was stored in password-protected computers, while consent and assent forms or any document containing identifying information were kept in lock and key cabinets that were only accessed by authorized study staff.

## Theoretical framework

The knowledge, attitudes, and practices (KAP) framework is a representative study of a specific population to collect information on what is known, believed, and done in relation to a particular topic [21,22]. KAP studies fundamentally assume a linear association between knowledge, attitude, and behavioral change [23] and in a qualitative study can highlight barriers and risks associated with requisite behavioral change (Fig 1).

## Data collection and analysis

The FGDs were conducted in Kiswahili in a space provided by the health facilities. Consent to record the discussions was obtained from all the participants. For privacy purposes, participants were assigned numbers as identifiers instead of using participants' names during the discussion.

A discussion guide was developed by the UNICEF MMAP initiative [24] and this was adapted for use in Kenya and covered the themes/ topics/ pre-determined questions looking at participants' understanding of mental health, caregiver response to adolescents in case of distress, adolescents' experiences with mental health conditions and its impact on their education, relationships, and family life. Additionally, the guide focused on participants' expectations from mental health professionals in case one could present with a mental health condition, what could be done, and their prior experiences handling mental health conditions.

The transcripts were uploaded on Qualitative Solution for Research (QSR) NVivo version 10 software. A thorough initial reading was done without coding to ensure adequate familiarization with the data set [25] as guided by thematic analysis principles. Thereafter, thematic content analysis was conducted. Data classification was done through line-by-line coding, and themes and sub-themes were identified using an integrated, inductive, and deductive analysis approach [26]. The priori/inductive themes were informed by the aims of the study. During the data analysis, matrix coding and queries were conducted for comparison purposes, identifying keywords frequencies and context of the coded statements, until saturation of data was observed, where no new themes emerged [20]. As a second independent step, the authors manually read through the transcripts and identified key themes, and the two sets of themes were merged. The manual coding helped in reviewing the placement of experiential understanding and knowledge of mental health problems including depression, anxiety, stress, and self-harm experienced by young people.

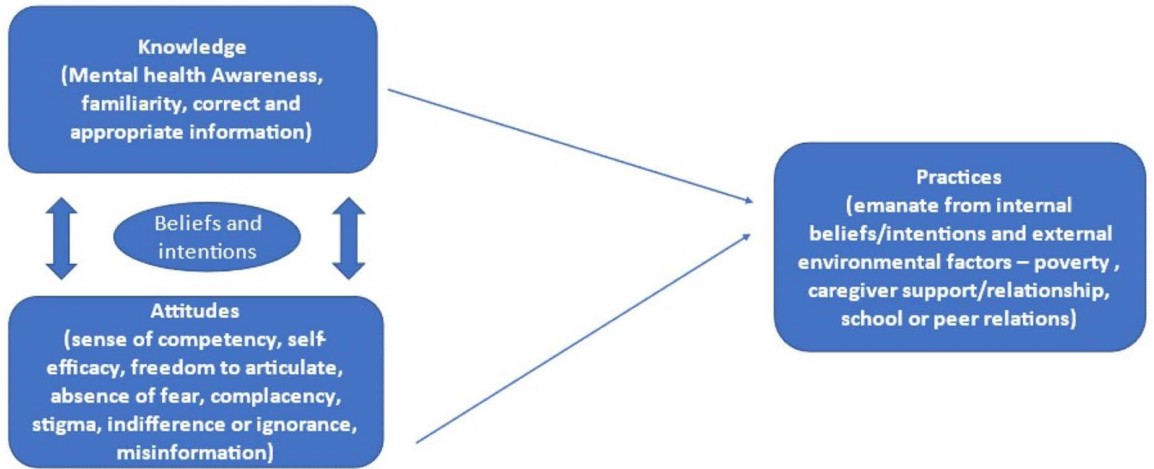

**Fig 1. Theoretical framework of knowledge, attitudes, and practices around mental health.**

## Results

### Demographic summary

About 47% were male adolescents and 53% were female adolescents. All adolescents spoke English and mainly Kiswahili, the two national languages, and often with a combination of other ethnic languages (See Table 2). Table 2 presents demographic information of adolescents and caregivers of adolescents between the age of 10–14 years who participated in the FGDs

### Overview of key findings

We identified six overarching themes that came up from the discussions in which knowledge, perceptions, and insights of adolescents' mental health were explored: a) Triggers of psychological disturbances, b) Attitudes towards mental health, c) Coping strategies used by adolescents to strengthen mental health, d) Barriers to strengthening mental health in young people, e) Effects of COVID-19 pandemic caregiver concerns around youth mental health and f) Community-based recommendations for mental health. Table 3 provides sub-themes associated with each of the themes to highlight associated ideas and concerns raised.

Also, see an illustrative image of how these themes are connected to various sub-themes in the NVivo-based analysis (Fig 2). Fig 2 connects the key themes to various sub-themes (micro-ideas, quotes that were repeated in response to a problem area explored in the interviews).

We now present consolidated findings about knowledge, attitudes, and practices around mental health that we found our participants echo in these intensive interviews.

**A. Knowledge about mental health/illnesses.** The adolescent respondents mainly described how to recognize someone with symptoms of mental health conditions that include loss of insight and hallucination, as illustrated in the following statement:

> *"He just goes absent-minded; he can just sit this way and goes absent-minded, then think of things that don't exist"* (boys ages 15-19 years)

However, caregivers highlighted some other subtle symptoms of psychological disturbances, such as social withdrawal and loss of concentration.

**Table 2. Adolescent and caregiver demographic information.**

|  | Category | Frequency | Percentage (%) |
|---|---|---|---|
| **Adolescents** | | | |
| Age (years) | 10–14 | 16 | 53.3 |
|  | 15–19 | 14 | 46.7 |
| Gender | Male | 14 | 46.7 |
|  | Female | 16 | 53.3 |
| Education level | Lower primary (class 3 and below) | 2 | 6.7 |
|  | Upper primary (class 4–8) | 16 | 53.3 |
|  | Secondary (form 1–4) | 10 | 33.3 |
|  | Post-secondary | 2 | 6.7 |
| **Caregivers** | | | |
| Age (years) | 30–39 | 6 | 37.5 |
|  | 40–49 | 9 | 56.3 |
|  | 50 and above | 1 | 6.2 |
| Gender | Male | 0 | 0 |
|  | Female | 16 | 100 |
| Education level | Primary | 3 | 18.7 |
|  | Secondary (form 1–4) | 13 | 81.3 |
|  | Post-secondary | 0 | 0 |

*"My understanding is maybe laziness in a child, sadness, sleeping, h/she doesn't want any work, sitting alone, doesn't want to eat, and some behaviors that I am not used to see in them" (caregivers of 10-14 years old)*

The respondents also described adolescents' experience with depression demonstrated by anger, refusal to talk (withdrawal), and feeling stressed.

*"If people have gone back to school- okay, your mother doesn't have money and so you are the only one who is at home, people are in school and when they come back they tell you, "School was fun today, this and that teacher," you start saying 'I wish I could have been there,' you start experiencing stress, you get depressed, you just become lonely" (15-19 years old girls)*

Substance abuse was reported as an indication of psychological disturbance in adolescents, and sometimes, it was also triggered by mental disturbances.

*"When he starts abusing drugs" (boys ages 15-19 years)*

The adolescents shared that suicidal behavior was associated with stressful home conditions (a place for protection turning out to be a stressful environment). The caregivers reported that suicidal behavior could be triggered by corporal punishment at home, and also child molestation or abuse were triggers to self-harm.

*"By the time the mother comes to know (that the child was molested), it will be too late, because the mum didn't create that good rapport with the child. So for you to notice that 'my child has been molested all along,' the time will have gone. So from all that you find that the questions we have been reading here appear on that child and so the final thing will be Suicidal" (caregivers of adolescents ages 10-14 years)*

**Table 3. Themes and sub-themes identified.**

| Themes | Subtithemes | Description |
|---|---|---|
| *Triggers of psychological disturbances* | • Traumatic experience like; Sexual abuse<br>• Dysfunctional family relationships- parents fighting, conflict between the adolescent child and the caregiver.<br>• Peer pressure<br>• Low self-esteem especially during the adolescence physical changes.<br>• Attaining poor grades in school<br>• Adolescent pregnancy and rejection by a sexual partner<br>• Economic challenges/poverty | Reported factors associated with psychological disturbance in adolescents |
| *Attitudes towards mental health/illness* | • Stigma<br>• Mistrust/Hiding distress<br>• Apathy/Neglect | Described as reaction towards persons with mental illness and also the language used to address them |
| *Coping Strategies used by adolescents* | • Engaging in outdoor activities to promote self-care<br>• Peer support<br>• Family support<br>• Seeking support from a professional | Respondents' description of the steps they take in case of psychological disturbances |
| *Barriers to strengthening mental health/ emerging concerns around youth mental health needs* | • Lack of community mental health services<br>• Stigma•   Inadequate knowledge on mental health | No counselors, no rehabilitation centers available in the community, high cost of seeking private professional counselors in urban centers |
| *Effects of COVID-19 pandemic and Emerging concerns around caregiver mental health and parenting* | • Economic challenges brought out by the effect of COVID-19 pandemic• Gender-based violence (Associated with increased psychosocial stressors due to the unexpected effects of the COVID-19 pandemic)<br>• Adolescents spending much time with mobile phones, watching television, and in play stations; a cause of conflict with parents (Especially during this period of Covid 19 pandemic) | Parent's economic challenges/poverty contribute to poor mental health of the children<br>(Situation worsened by loss of livelihoods during this time of COVID-19 pandemic).<br>COVID-19 pandemic caused interruption of school calendar, and children had to spend much time idle at home (Also with idle parents at home) |
| *Community recommendation on interventions* | • Prioritize persons with substance use disorders<br>• Address stigma<br>• Increase access to mental health services - Free mental health clinic<br>• Address knowledge gap; create awareness<br>• Role of religious institutions; encourage adolescents to participate in religious activities | Participants reported prioritization of adolescent substance use disorder, stigma, increased mental health knowledge, and youth involvement in religious activities. |

i. Mental health knowledge gaps

Participants reported several mental health knowledge gaps, which led to late diagnosis when the disease had already progressed to severe mental disorder.

*"Matters of mental health according to me I think it is supposed to be spread through a lot of education and awareness because you find that not many people in the community have information on mental health; we notice that maybe somebody has a mental health problem after it has gone overboard, maybe amefyatuka (become insane) …`" (caregivers of adolescents ages 10-14 years)*

Younger adolescents had less experience and knowledge than the older ones. One of their responses was that surgery was a treatment for severe mental illness.

*"(When a person has a mental illness) He goes through head surgery to establish the problem" (boys ages 10-14 years)*

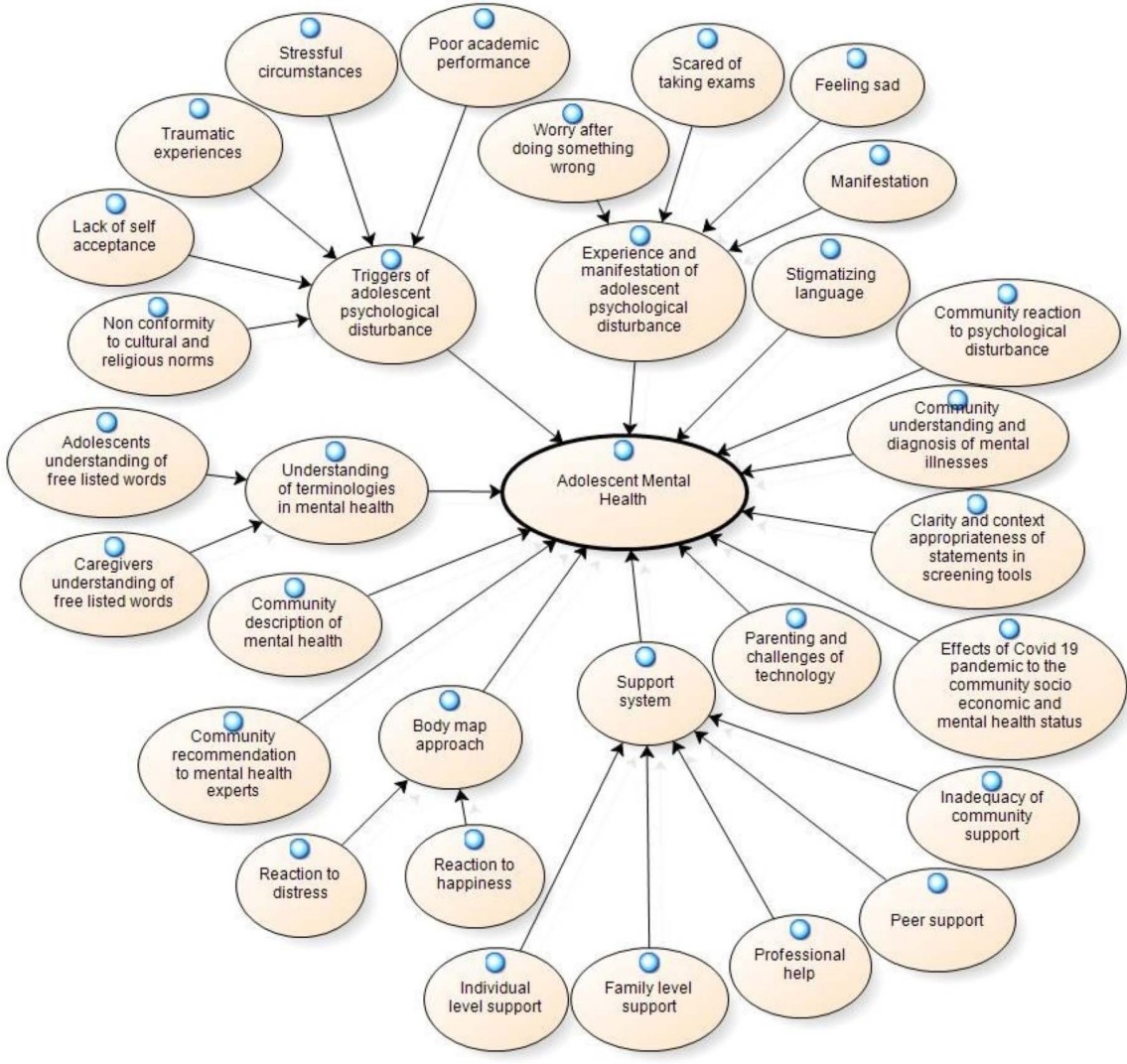

**Fig 2. Thematic nodes and sub-nodes.**

**B. Triggers of psychological disturbance.** The respondents reported various causes of adolescent psychological disturbance, including:

i. Traumatic experiences like sexual abuse and maltreatment were identified to trigger mental illnesses.

*"A child who has gone through abuse; let's say like rape, so it means that the child will be restless due to experiencing the abuse"* (caregivers of 10-14 years old)

ii. Dysfunctional family relationships - parents fighting, conflict between adolescent and caregiver.

Family disturbances and poor relationships were known to be triggers of distress, especially depression and anxiety.

*"If your father and mother always fight or if your father has ever wanted to- what is it called? To take advantage of you, you can get stressed because obviously you cannot share with outsider and there is a way in which you will be afraid of telling your mother, so you will just keep it to yourself"* (girls ages 15-19 years).

iii. Peer pressure

We identified inappropriate peer pressures to be the source of mental disturbances and high-risk behaviors.

*"Because they have reached adolescence, they are at the puberty level; if she is a girl she wants to know things to do with boyfriends, she feels that she has developed breasts; her body has changed and starts being shy and many other things"* (caregivers of adolescents ages 10-14 years)

iv. Pressure to achieve high grades in school was identified as a source of stress for young people. It was also reported as a reason for harsh disciplining at home.

*"Sometimes at school you have best friends and when it comes to exams these best friends are passing, when results come out you find that they are celebrating but you have failed and you start asking yourself, 'why me?' So in those emotions is where stress hits you"* (girls ages 15-19 years)

v. Adolescent pregnancy and rejection by sexual partner were causes of distress to young girls.

*"I can say that you can get stressed psychologically for example after betrayal; if I use a real example, the way she had said before that you could have a boyfriend responsible for your pregnancy by mistake then he denies and refuses and says that he is not responsible, so you know in that way you have been betrayed and that thing must disturb your mind"* (girls ages 15-19 years)

vi. Economic challenges/poverty were the primary reasons for the experience

*"When we didn't have money we were just living a hard life, we were not getting something to eat, so I used to see some going in their house and cooking- as in we used to see others cooking"* (girls ages 15-19 years)

**C. Attitudes towards mental health/illness.**

i. Internalized and externalized Stigma

The use of stigmatizing language was identified in the respondent's description of persons with mental illnesses. This was a significant barrier in enabling young people and their families to seek support when in need.

*"They know the name they will say, 'ule boy mchizi,' (that insane boy) that is what they say"* (boys ages 10-14 years)

The social isolation that ensues makes people who experience distress feel really discriminated and alone.

*"When some see such people (with mental illness) they get out of the way and reject them, so some feels as if h/she is nothing in the world"* (caregivers of adolescents ages 10-14 years)

ii. Apathy/Neglect

Our respondents shared that the "don't care" kind of attitude towards persons with mental illnesses leads to neglect and apathy towards those in mental distress.

*"People are different; there are those who will run away from you" (boys ages 15-19 years)*

iii. Mistrust between caregivers and adolescent

The participants reported that mistrust could hinder the adolescents' sharing of distressing feelings and prevent them from opening up to their caregivers.

*"People fail to talk about it because they wonder- if I tell you my problems how will you perceive me? As in I will not be able to face you because I cannot tell how you perceived me after telling you those things, I don't know the person you will tell, okay; there is no trust as such" (girls ages 15-19 years)*

Adolescents also felt they had to hold back, knowing they would not be understood. Similarly, caregivers said they knew their child was not ready to share and held back information.

*"Sometimes you can talk to a child and h/she just stares at you this way as if h/she doesn't hear you, you talk again but h/she just looks at you, you speak to him/her for the third time, but h/she just looks at you; so you see? That child is not open" (caregivers of adolescents ages 10-14 years)*

**D. Coping strategies by young people and caregivers in strengthening mental health.**

  i. Outdoor activities to promote self-care were identified as a means to promote happiness and build oneself up. We saw that this could be a gendered response, but young people sought sports and leisure as the way out of stress and strains.

  **"By engaging in activities that keep me busy …. Playing football, dancing" (boys ages 15-19 years)*

Peer and family support were highlighted as strategies to improve coping and garner support.

  **"Communication; talking to the child and understanding him/her, then we should stop being harsh to the children, we should be friendly with your child then h/she will not hide anything from you" (caregivers of adolescents ages 10-14 years).*

ii. Referral for professional help also came up as a potential source of support for both boys and girls in our FGDS.

  *"You can look for a counselor" (caregivers of adolescents ages 10-14 years).*

**Emerging concerns around adolescent mental health needs**

The lack of community mental health services emerged as a significant barrier to providing timely support to the adolescent.

  *"This is our community at (name withheld) and there is not even a rehabilitation, there is no counselor and if it is about a counselor most of them you go and pay, you know? So there is no support in the community totally" (caregivers of adolescents ages 10-14 years)*

Knowledge gaps and stigma that hinder caregivers from responding appropriately to psychological disturbances experienced by adolescents were repeatedly articulated. Here is an example of what caregivers shared.

  *"There are other parents who are not understanding; yes, you could be having stress and yes, there is a way in which you will behave at home, before your mother wants to know why you are that way she will start to tell you, "you are*

*bringing me your problems, yet I have my own problems? I also have my stresses that are disturbing me, do not bring me your stresses" (boys ages 15-19 years)*

An increase in mental health knowledge and literacy would likely help in addressing mental health stigma. This will enable early diagnosis of mental illness and hence improve access to interventions, as discussed in our findings [27].

### Emerging concerns around caregiver mental health and parenting

Caregivers articulated that the excessive use of technology, especially mobile phones and PlayStations, negatively affects parent-child communication and, consequently, the mental health of adolescents who spend too much time on gadgets instead of engaging in domestic chores and relaxation activities. Our participants also shared that caregivers can easily become stressed and feel guilty (with self-blame) about the deteriorating mental health of the adolescent. The following quotes from the caregivers are a testament to caregiver challenges.

*"They (parents) feel guilty because they did not protect him when that thing (psychological disturbance) happened" (boys ages 10-14 years)*

The COVID-19 pandemic brought unprecedented economic challenges that affected the livelihoods of families, which had a negative spill-over on the caregivers' and adolescents' mental health. According to the caregivers, these economic challenges contributed to increased Gender-Based Violence (GBV) and adolescent pregnancies.

*"(COVID-19 pandemic) It has also contributed to these GBV issues; people start- there are these fights between father and mother sometimes because- but probably it is due to lack of money" (caregivers of adolescents ages 10-14 years)*

## Discussion

The findings illustrated that adolescents experience significant psychological pressures and disturbances in their every-day lives, and there are familial, peer, and individual level triggers of these problems, as also reported in other studies [28–31]. These disturbances disrupt learning and significantly impair the social functioning of the affected adolescent. The disturbed adolescent could manifest rebelliousness, substance use, social withdrawal, suicidal behavior, and deteriorating school performance as a result of the mental distress experienced.

In response to their knowledge about mental illnesses, adolescent participants mostly described overt mental health conditions. The respondents identified triggers of mental health conditions as sexual abuse, poverty, economic challenges, dysfunctional family relationships such as parents fighting, and misunderstandings between parent and child. Additional triggers that were reported were negative peer pressure, low self-esteem, especially during the period of adolescent physical changes, and attaining poor grades in school. Caregivers demonstrated knowledge of subtle symptoms of psychological disturbances.

Our findings corroborate with evidence from other research studies that have also associated mental illness with increasing poverty brought out by severe economic challenges [32,33] in resource-constrained households. In terms of their attitudes, stigma towards persons with mental disorders was identified as a key problem that caregivers and adolescents struggled with, and this includes the use of stigmatizing language in describing persons with mental illnesses. This is a hindrance to seeking intervention in the community and with young people themselves.

In response to our questions about practices and behaviors towards individuals, especially adolescents and young people with psychological disturbances, our caregivers pointed out that parents feel guilty that they failed to protect their child when it is the youth who becomes unwell. We found that community members seek out spiritual interventions through prayers, and some community members demonstrate apathy towards persons with mental disorders. Young participants were very sensitive about this and mentioned that they think greater support and positive attitudes need to be shown towards vulnerable individuals living with mental illnesses.

## Caregiver mental health is a predictor of offspring's physical and mental health

A study carried out in Kangemi and Kariobangi informal settlements found that adverse childhood experiences (ACEs) were highly prevalent [34]. In tandem with cross-cultural evidence on the family stress model, this study found that maternal ACEs are robust predictors for poor child mental health. These findings also apply to adolescents. Adolescents' experiences of adversity and sexual or physical abuse have been associated with increased vulnerable attachment toward parents/caregivers. Poor socioeconomic status was also associated with higher problem behavior development as well as an insecure attachment with caregivers [31]. Secure attachment, among others, was also reported as a protective factor for high-risk behaviors and the development of mental health problems later in life [7].

## Limitations

We conducted this study during the COVID-19 pandemic, so we could not freely and closely interact with our participants and observe their nonverbal expressions. We were also unable to obtain male caregivers; therefore, their views were not represented.

## Conclusion

Psychosocial disturbance significantly affects the mental well-being of adolescents and impairs their performance of daily routine functions. This has far-reaching negative consequences for adolescents and caregivers, including violence, depression, and suicidal behavior. Therefore, it is imperative to develop sustainable community interventions that loop in the caregivers and all other key stakeholders to help reduce harmful exposures, strengthen resilience to recover from exposure to adverse experiences, and improve coping with everyday stressors. For instance, faith-based organizations and primary health centers can play a crucial role in promoting adolescent and community mental health by bridging gaps in mental health knowledge, reducing stigma, and increasing access to appropriate youth-friendly psychosocial interventions. In addition, the COVID-19 pandemic had significant negative effects on the mental well-being of both adolescents and caregivers. Therefore, community mental health programs should address contemporary issues that are potential triggers of psychological disturbance. In the current context, caregiver and adolescent mental health needs to be addressed through prevention and mental health promotion campaigns. Community-based support would ensure that individuals maintain some resilience through this challenging period and build themselves better now and in the post-pandemic world.

## Acknowledgments

Authors would like to thank other mentors in INSPIRE Kenya work, adolescents and their caregivers who participated, and a fantastic team of community health workers and health facility workers of Kariobangi and Kangemi for their support.

## Author contributions

**Conceptualization:** Brandon Kohrt, Liliana Carvajal Velez, Manasi Kumar.

**Data curation:** Darius Nyamai.

**Formal analysis:** Darius Nyamai, Manasi Kumar.

**Funding acquisition:** Priscilla Idele, Liliana Carvajal Velez, Manasi Kumar.

**Investigation:** Obadia Yator, Ian Kanyanya, Bruce Chorpita, Liliana Carvajal Velez, Manasi Kumar.

**Methodology:** Vincent Nyongesa, Bruce Chorpita, Beatrice Madeghe, Liliana Carvajal Velez, Manasi Kumar.

**Project administration:** Vincent Nyongesa, Joseph Kathono, Shillah Mwavua, Jill Ahs, Beatrice Madeghe, Priscilla Idele, Liliana Carvajal Velez.

**Resources:** Manasi Kumar.

**Software:** Darius Nyamai.

**Supervision:** Vincent Nyongesa, Joseph Kathono, Shillah Mwavua, Obadia Yator, Ian Kanyanya, Nabila Amin, Simon Njuguna, Jill Ahs, Brandon Kohrt, Beatrice Madeghe, Priscilla Idele.

**Validation:** Vincent Nyongesa, Joseph Kathono, Shillah Mwavua, Obadia Yator, Nabila Amin, Manasi Kumar.

**Visualization:** Joseph Kathono, Manasi Kumar.

**Writing – original draft:** Vincent Nyongesa, Manasi Kumar.

**Writing – review & editing:** Vincent Nyongesa, Joseph Kathono, Shillah Mwavua, Darius Nyamai, Obadia Yator, Ian Kanyanya, Nabila Amin, Simon Njuguna, Jill Ahs, Brandon Kohrt, Bruce Chorpita, Beatrice Madeghe, Priscilla Idele, Liliana Carvajal Velez.

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
