## [Decision Letter · Decision Letter 0]

18 Mar 2025

PONE-D-24-44355“If I tell you my problems, how will you perceive me?”: A Qualitative Appraisal of Mental Health Knowledge, Barriers, and Opportunities for Care among Kenyan AdolescentsPLOS ONE

Dear Dr. Kumar,

Thank you for submitting your manuscript to PLOS ONE. After careful consideration, we feel that it has merit but does not fully meet PLOS ONE’s publication criteria as it currently stands. Therefore, we invite you to submit a revised version of the manuscript that addresses the points raised during the review process.

We look forward to receiving your revised manuscript.

Kind regards,

Stefaan Six, Ph.D.

Academic Editor

PLOS ONE

Journal Requirements:

“Research reported in this publication was supported by the Fogarty International Center of the National Institutes of Health under Award Number K43TW010716, which also supported the contributions of MK to this work. The content is solely the responsibility of the authors and does not necessarily represent the official views of the National Institutes of Health. The qualitative inquiry is part of an embedded study protocol for the Measurement of Mental Health of Adolescents at population level (MMAP) led by UNICEF and the inquiry reported here was partially supported by UNICEF Innocenti office.”

4. Please note that funding information should not appear in the Acknowledgments section or other areas of your manuscript. We will only publish funding information present in the Funding Statement section of the online submission form. Please remove any funding-related text from the manuscript. 

5. We note that you have indicated that there are restrictions to data sharing for this study. For studies involving human research participant data or other sensitive data, we encourage authors to share de-identified or anonymized data. However, when data cannot be publicly shared for ethical reasons, we allow authors to make their data sets available upon request. For information on unacceptable data access restrictions, please see http://journals.plos.org/plosone/s/data-availability#loc-unacceptable-data-access-restrictions. 

6. In this instance it seems there may be acceptable restrictions in place that prevent the public sharing of your minimal data. However, in line with our goal of ensuring long-term data availability to all interested researchers, PLOS’ Data Policy states that authors cannot be the sole named individuals responsible for ensuring data access (http://journals.plos.org/plosone/s/data-availability#loc-acceptable-data-sharing-methods).

7. When completing the data availability statement of the submission form, you indicated that you will make your data available on acceptance. We strongly recommend all authors decide on a data sharing plan before acceptance, as the process can be lengthy and hold up publication timelines. Please note that, though access restrictions are acceptable now, your entire data will need to be made freely accessible if your manuscript is accepted for publication. This policy applies to all data except where public deposition would breach compliance with the protocol approved by your research ethics board. If you are unable to adhere to our open data policy, please kindly revise your statement to explain your reasoning and we will seek the editor's input on an exemption. Please be assured that, once you have provided your new statement, the assessment of your exemption will not hold up the peer review process.

8. Your ethics statement should only appear in the Methods section of your manuscript. If your ethics statement is written in any section besides the Methods, please delete it from any other section. 

9. Please include a separate caption for each figure in your manuscript.

Reviewers' comments:

Reviewer's Responses to Questions

**Comments to the Author**

1. Is the manuscript technically sound, and do the data support the conclusions?

Reviewer #1: Yes

Reviewer #2: Yes

2. Has the statistical analysis been performed appropriately and rigorously? 

Reviewer #1: I Don't Know

Reviewer #2: Yes

3. Have the authors made all data underlying the findings in their manuscript fully available?

Reviewer #1: Yes

Reviewer #2: Yes

4. Is the manuscript presented in an intelligible fashion and written in standard English?

Reviewer #1: Yes

Reviewer #2: Yes

5. Review Comments to the Author

Reviewer #1: It is an interesting topic; "If I tell you my problems, how will you perceive me?”: A Qualitative Appraisal of Mental Health Knowledge, Barriers, and Opportunities for Care among Kenyan adolescent's ."

1. Abstract:- please remove the discssion part from the abstract section. No need of mentioning it in the abstract.

2. Methodology:- You have used a qualitative methods, but please specify which type of qualitative study design/method you have used in your study?

3. Ethical clearance Vs Ethical approval:- Please not to write the ethical clearance in two sections, i.e remove the ethical clearance secthon. Additionaly, there are two different ethical approval number; Kenyan national commission for Science, Technology and innovation NACOSTI/P/19 and NACOSTI/P/8757. Which one is your correct approval No.??

Reviewer #2: Reviewer comment and suggestions

Generally congratulation to the authors for the interesting paper which informed the health of adolescent and their caregiver however there is several issues need to improve.

Adhere to journal guideline in organizing the work

Title

• Your title is understandable but could be clearer and more structured also qualitative appraisal is fine but specifying ‘’A QUALITATIVE STUDY ‘’makes it more formal.

Abstract

• On this part the background/introduction it’s very narrow the authors should revise and improve and I noted the (LMICs) the ways explain its not formal revise.

• There is no need to write the discussion on part of abstract

• I would ask the authors how many population in this study?

INTRODUCTION

• I did not understand COVID-19 come from the author should be more specific the reader will confuse revise and start from the title.

METHOD /MATERIAL

• I noted on part of method and material the authors write (QUALITATIVE METHODOLOGY) revise and improve

• How trustworthiness was assured?

DISCUSSION

• Well written but the authors should remove this word (Overall summary findings)

Reference

• Several references do not fit the requirements of Vancouver style. Revise and improve them.

6. PLOS authors have the option to publish the peer review history of their article (what does this mean? ). If published, this will include your full peer review and any attached files.

**Do you want your identity to be public for this peer review?** For information about this choice, including consent withdrawal, please see our Privacy Policy .

Reviewer #1: **Yes: ** Agmas Wassie Abate

Reviewer #2: **Yes: ** rehema abdallah

---

## [Author Response · Author response to Decision Letter 1]

9 Apr 2025

31st March 2025

To

The Editor

PLOS ONE

Dear Editor,

Re: Resubmission of paper PONE-D-24-44355 “If I tell you my problems, how will you perceive me?”: A Qualitative Appraisal of Mental Health Knowledge, Barriers, and Opportunities for Care among Kenyan Adolescents

We want to thank you for reviewing our manuscript. We are grateful to the reviewers for their comments and feedback. Below is a point-by-point response.

We hope the edited paper will meet your expectations.

Regards

Response: Thank you, we have done this

Response:

“Research reported in this publication was supported by the Fogarty International Center of the National Institutes of Health under Award Number K43TW010716, which also supported the contributions of MK to this work. The content is solely the responsibility of the authors and does not necessarily represent the official views of the National Institutes of Health. The qualitative inquiry is part of an embedded study protocol for the Measurement of Mental Health of Adolescents at population level (MMAP) led by UNICEF and the inquiry reported here was partially supported by UNICEF Innocenti office.”

Response:

4. Please note that funding information should not appear in the Acknowledgments section or other areas of your manuscript. We will only publish funding information present in the Funding Statement section of the online submission form. Please remove any funding-related text from the manuscript.

Response:

5. We note that you have indicated that there are restrictions to data sharing for this study. For studies involving human research participant data or other sensitive data, we encourage authors to share de-identified or anonymized data. However, when data cannot be publicly shared for ethical reasons, we allow authors to make their data sets available upon request. For information on unacceptable data access restrictions, please see http://journals.plos.org/plosone/s/data-availability#loc-unacceptable-data-access-restrictions.

Response: There are no ethical or legal restrictions on sharing data set. Our study received ethical approval from Kenyatta National Hospital-University of Nairobi Ethics review committee (https://erc.uonbi.ac.ke/ ), whose contact is uonknh_erc@uonbi.ac.ke

Response: Thank you, we have uploaded our anonymized data as Supporting information file.

Response:

6. In this instance it seems there may be acceptable restrictions in place that prevent the public sharing of your minimal data. However, in line with our goal of ensuring long-term data availability to all interested researchers, PLOS’ Data Policy states that authors cannot be the sole named individuals responsible for ensuring data access (http://journals.plos.org/plosone/s/data-availability#loc-acceptable-data-sharing-methods).

Response: Our study received ethical approval from Kenyatta National Hospital-University of Nairobi Ethics review committee (https://erc.uonbi.ac.ke/ ), who can be contacted on email via uonknh_erc@uonbi.ac.ke

7. When completing the data availability statement of the submission form, you indicated that you will make your data available on acceptance. We strongly recommend all authors decide on a data sharing plan before acceptance, as the process can be lengthy and hold up publication timelines. Please note that, though access restrictions are acceptable now, your entire data will need to be made freely accessible if your manuscript is accepted for publication. This policy applies to all data except where public deposition would breach compliance with the protocol approved by your research ethics board. If you are unable to adhere to our open data policy, please kindly revise your statement to explain your reasoning and we will seek the editor's input on an exemption. Please be assured that, once you have provided your new statement, the assessment of your exemption will not hold up the peer review process.

Response: this is a qualitative data which can potentially be used to identify participants if several alternations are not made. We prefer not to share interview transcripts open-access but the senior author can provide upon reasonable request when contacted via email.

8. Your ethics statement should only appear in the Methods section of your manuscript. If your ethics statement is written in any section besides the Methods, please delete it from any other section.

Response: Thank you, we have deleted it from elsewhere and retained it in the methods section only

9. Please include a separate caption for each figure in your manuscript.

Response: Thank you, we have included figure captions (please see pages 5 and 8)

Response: Thank you, we have reviewed the references

Review Comments to the Author

Reviewer #1: It is an interesting topic; "If I tell you my problems, how will you perceive me?”: A Qualitative Appraisal of Mental Health Knowledge, Barriers, and Opportunities for Care among Kenyan adolescent's ."

1. Abstract:- please remove the discssion part from the abstract section. No need of mentioning it in the abstract.

Response: Thank you for this suggestion, we have removed the discussion from the abstract

2. Methodology:- You have used a qualitative methods, but please specify which type of qualitative study design/method you have used in your study?

Response: this is an anthropologically embedded cultural validation of mental health problems and measurement tools study. Such studies aim to address culturally responsive approaches to addressing measurement of mental health problems.

3. Ethical clearance Vs Ethical approval:- Please not to write the ethical clearance in two sections, i.e remove the ethical clearance secthon. Additionaly, there are two different ethical approval number; Kenyan national commission for Science, Technology and innovation NACOSTI/P/19 and NACOSTI/P/8757. Which one is your correct approval No.??

Response: We have removed the section of ethical clearance and replaced it with ethical approval. We apologize for the oversight, the correct NACOSTI number is NACOSTI/P/19/77705/28063

Reviewer #2: Reviewer comment and suggestions

Generally congratulation to the authors for the interesting paper which informed the health of adolescent and their caregiver however there is several issues need to improve.

Adhere to journal guideline in organizing the work

Title

• Your title is understandable but could be clearer and more structured also qualitative appraisal is fine but specifying ‘’A QUALITATIVE STUDY ‘’makes it more formal.

Response: Thank you for pointing to this, we have now revised it to “A Qualitative Study” per your suggestion

Abstract

• On this part the background/introduction it’s very narrow the authors should revise and improve and I noted the (LMICs) the ways explain its not formal revise.

Response:

• There is no need to write the discussion on part of abstract

Response: Thank you, we have removed the discussion part from the abstract

• I would ask the authors how many population in this study?

Response: Thank you for pointing this out; however, in the materials and methods section of the abstract, we recorded that our study population was 46 participants, consisting of adolescents and their caregivers.

INTRODUCTION

• I did not understand COVID-19 come from the author should be more specific the reader will confuse revise and start from the title.

Response: Thank you, we have now included COVID-19 in the title per your suggestion to make it clear

METHOD /MATERIAL

• I noted on part of method and material the authors write (QUALITATIVE METHODOLOGY) revise and improve

Response: we have made tweaks to the sections making it more specific.

• How trustworthiness was assured?

Response: For confidentiality and trustworthiness, the participants were given identifying numbers, and did not say their names throughout the FGDs. Recorded FGDs and transcripts were saved in password-protected computers, accessible to research team members only. In this way, no participant's identifying information was accessed by anyone outside the research team.

DISCUSSION

• Well written but the authors should remove this word (Overall summary findings)

Response: Thank you for this suggestion, we have now removed the sub-heading

Reference

• Several references do not fit the requirements of Vancouver style. Revise and improve them.

Response: We have now revised the references and improved them

---

## [Editor Report · Decision Letter 1]

16 Apr 2025

“If I tell you my problems, how will you perceive me?”: A Qualitative Study of Mental Health Knowledge, Barriers, and Opportunities for Care among Kenyan Adolescents During COVID-19

PONE-D-24-44355R1

Dear Dr. Kumar,

We’re pleased to inform you that your manuscript has been judged scientifically suitable for publication and will be formally accepted for publication once it meets all outstanding technical requirements.

Kind regards,

Stefaan Six, Ph.D.

Academic Editor

PLOS ONE
---

## [Editor Report · Acceptance letter]

PONE-D-24-44355R1

PLOS ONE

Dear Dr. Kumar,

I'm pleased to inform you that your manuscript has been deemed suitable for publication in PLOS ONE. Congratulations! Your manuscript is now being handed over to our production team.

Kind regards,

on behalf of

Dr. Stefaan Six

Academic Editor

PLOS ONE